# Prevalence of underlying diseases in died cases of COVID-19: A systematic review and meta-analysis

Fatemeh Javanmardi[1], Abdolkhalegh Keshavarzi[2], Ali Akbari[3], Amir Emami[1]*, Neda Pirbonyeh[1]

1 Microbiology Department, Burn and Wound Healing Research Center, Shiraz University of Medical Sciences, Shiraz, Iran, 2 Surgery Department, General Surgery, Burn and Wound Healing Research Center, Shiraz University of Medical Sciences, Shiraz, Iran, 3 Department of Anesthesiology, School of Medicine, Shiraz University of Medical Sciences, Shiraz, Iran

* Emami.microbia@gmail.com

**Data Availability Statement:** All relevant data are within the paper and its Supporting Information files.

## Abstract

### Introduction

Underlying disease have a critical role in vulnerability of populations for a greater morbidity and mortality when they suffer from COVID-19. The aim of current study is evaluating the prevalence of underlying disease in died people with COVID-19.

### Methods

The current study have been conducted according to PRISMA guideline. International database including PubMed, Scopus, Web of Science, Cochrane and google scholar were searched for relevant studies up to 1 June. All relevant articles that reported underlying disease in died cases of COVID-19 were included in the analysis.

### Results

After screening and excluding duplicated and irrelevant studies, 32 articles included in the analysis. The most prevalent comorbidities were hypertension, diabetes, cardiovascular disease, liver disease, lung disease, malignancy, cerebrovascular disease, COPD and asthma. Among all reported underlying disease, highest and lowest prevalence was related to hypertension and asthma which were estimated 46% (37% - 55%) and 3% (2%- 6%), respectively.

### Conclusion

In summary, underlying disease have a critical role in poor outcomes, severity of disease and high mortality rate of COVID-19 cases. Patients with hypertension, cardiovascular disease and diabetes should be carefully monitored and be aware of health protocols.

**Funding:** No, we had not received any fees and fund

**Competing interests:** The authors have declared that no competing interests exist.

## Introduction

The year 2020 began with a global pandemic, caused by SARS-CoV-2, a highly contagious novel virus which lead a big health challenge in the world. During less than 9 months, COVID-19 influence on more than 20,000,000 million people and causes 740,000 deaths till now (12 August) [1]. Although majority of cases are in mild and moderate and even with no symptoms, but for some infected individuals, the incidence of disease is along with serious complications such as severe pneumonia, acute respiratory distress, multi organ failure and finally death [2]. Nowadays a well-known reason for death is SARS-CoV-2, although there is no accurate information about mortality rate; especially it varies in different countries and reported between 1.4% to 4.3% [3]. According to literature studies, the basic reason for death related to COVID-19 was introduced pneumonia; more over it was found that pre-existing morbidities are significantly increase the related mortality rate [4]. Another important factor increasing the risk of mortality in this crisis is the distance between the incidence of disease and hospitalization. According to different reports, time range of symptoms progress in COVID-19 death is between 6 to 41 days [5]. Further comparative analysis have been revealed that sever form of SARS-CoV-2 may appear in older people which is similar to other respiratory infections such as influenza, severe acute respiratory syndrome (SARS), and Middle East respiratory syndrome (MERS) [6]. Since SARS-CoV-2 is a novel virus, little information and majority of uncertainties are present about mechanism of disease which is created a serious threat for infected cases. Basic analysis about the cause of death have shown critical evidences about the impact of underlying diseases on death related to COVID-19 [4, 7]. In fact, these patients have been identified as particularly vulnerable populations for a greater morbidity and mortality when they suffer from COVID-19. In the current emergency outbreak related to COVID-19, due to the majority of uncertain data around the world, and the lack of certain treatment and vaccine for this infection, it is the time of investigation and research. Based on evidences, this information will undoubtedly be key to the knowledge and control of mortality in times of this pandemic. So, the aim of current study is designed to evaluating the prevalence of underlying diseases in died people with COVID-19.

## Methods

### Search strategy

The current study have been conducted according to Preferred Reporting Items for Systematic Reviews and Meta-Analyses (PRISMA) guideline and is registered in PROSPERO (CRD42020186617) in 29 June 2020.

International database including PubMed, Scopus, Web of Science, Cochrane and google scholar were searched for relevant studies up to first June. Search strategy were done based on Mesh keywords as follow: "fatality AND COVID-19", OR "died AND COVID-19", OR "death AND COVID-19", "deceased AND COVID-19" and "mortality AND COVID-19". S1 Table in S1 File is provided the Mesh terms in detail. Further evaluation was carried out in reference of proper articles for more papers. Search terms were restricted to English language, but due to high number of articles in Chinese language, abstracted were assessed in these studies.

### Inclusion and exclusion criteria

Two authors (F.J and A.E) independently evaluated the studies and in case of disagreement the third author decide about it. Included criteria were defined as follow: any articles about death related to COVID-19, studies which reported underlying diseases in died patients. Articles in

preprint status and with inappropriate information were excluded from the analysis process. Quality assessment were conducted by Newcastle Ottawa Scale and the related results have been provided in S2 Table in S1 File [8]. Moreover, characteristics of included studies are shown in Table 1.

## Statistical analysis

Pooled prevalence with 95% confidence Interval were estimated by applying inverse-variance weighted method. Evaluation for heterogeneity was done based on Higgins $I^2$ and Cochrane Q statistics. Heterogeneity was defined as low ($I^2 < 25\%$), high ($I^2 > 50\%$) and moderate (25–50%). In case of high heterogeneity, random effect model was used. Publication bias were assessed by funnel plot and Egger's test. Statistical analysis was conducted by STATA 13. P-value less than 0.05 was considered statistically significant.

**Table 1. Characteristics of included studied in meta-analysis.**

| Authors | Number of Death | Hypertension | Diabetes | Heart diseases | Kidney diseases | COPD | Malignancy | Liver disease | lung Disease | Cerebrovascular |
|---|---|---|---|---|---|---|---|---|---|---|
| Fan Yang, et al [9] | 92 | 51 | 13 | 16 | 2 | 1 | 4 | 3 | | 10 |
| Qiurong Ruan, et al [10] | 68 | | | 36 | | | | | | |
| Mark M. Alipio, et al [11] | 50 | 34 | 23 | | 17 | | | | | |
| Jianfeng Xie, et al [12] | 168 | 84 | 42 | 31 | | | | | 16 | |
| Wei-jie Guan, et al [13] | 50 | | | | | 6 | | | | 6 |
| Francesco Violi, et al [14] | 64 | 39 | 16 | 13 | | 14 | | | | |
| Amir Emami, et al [15] | 87 | 17 | 27 | 23 | 8 | 4 | 10 | | | |
| Reza Shahriarirad, et al [16] | 9 | 2 | 2 | 2 | | 1 | | | | |
| Mohammad Nikpouraghadam, et al [17] | 239 | 8 | 11 | 4 | 3 | | 1 | | | |
| Yan Deng, et al [18] | 109 | 40 | 17 | 13 | | | 6 | | 22 | |
| Yongli Yan, et al [19] | 108 | 57 | 39 | 27 | 1 | | | | | |
| Graziano Onder, et al [20] | 355 | | 126 | 117 | | | 72 | | | |
| Marcello Covino, et al [21] | 23 | 10 | 2 | | | 4 | 1 | | | 9 |
| Xun Li1, et al [22] | 25 | 16 | 10 | 8 | 5 | 2 | 2 | | | 4 |
| Mingli Yuan, et al [23] | 10 | 5 | 6 | 3 | | | | | | 1 |
| Fei Zhou, et al [24] | 54 | 26 | 17 | 13 | 2 | | | | 4 | |
| Jianlei Cao, et al [25] | 17 | 11 | 6 | 3 | 3 | | 1 | 1 | | 3 |
| Jianbo Tian, et al [26] | 46 | | | | | | 46 | | | |
| Lang Wang, et al [27] | 65 | 32 | 11 | 21 | 4 | 11 | 3 | 1 | | 10 |
| Chaomin Wu, et al [28] | 44 | 16 | 11 | 4 | | | | | | |
| Rong-Hui Du, et al [29] | 21 | 13 | 6 | 12 | | | 1 | | | |
| Bicheng Zhang, et al [30] | 82 | 46 | 15 | 17 | 4 | 12 | 6 | 2 | | 10 |
| Kunyu Yang, et al [31] | 40 | 11 | 2 | 5 | | | | | | |
| Yingzhen Du, et al [32] | 85 | 58 | 32 | 10 | 3 | 2 | 6 | 7 | | 7 |
| Chaomin Wu, et al [33] | 44 | 16 | 11 | 4 | | | | | | 5 |
| CDC Korea [34] | 66 | 30 | 23 | 10 | 5 | | 7 | | | |
| Yifei Chen, et al [35] | 38 | 15 | 11 | 4 | | 1 | 1 | | 3 | |
| Ya-Jun Sun, et al [36] | 100 | 41 | 29 | 27 | | | | | | 12 |
| Junli Li, et al [37] | 14 | 10 | 3 | 4 | | | | | | |
| Yiguang Chen, et al [38] | 50 | 17 | 13 | | | | | | | |
| Lei Chen, et al [39] | 208 | 104 | 59 | 63 | 20 | 12 | 17 | | | |

## Results

According to initial search, total of 6507 articles were found in different databases. After screening and excluding duplicated and irrelevant studies, finally 32 articles met the inclusion criteria and considered in the analysis (Fig 1).

Through the current meta-analysis, 28 studies reported the incidence of COVID-19 in hypertensive patients. Among all reported underlying diseases, highest prevalence was related to hypertension which was estimated 46% (37% - 55%) (Fig 2). Significant and high heterogeneity was observed between studies ($I^2$ = 92.84%, P<0.001). Although publication bias was observed based on funnel plot in S1 Fig in S2 File and Egger test (t = 3.19, p = 0.004).

The overall prevalence of diabetic comorbidities estimated 26% (21%-31%) in fatal cases of COVID-19 (Fig 3). High and significant heterogeneity between 29 studies cause to use random effect model. Egger's test is indicating publication bias (t = 4.45, P = 0.001), also funnel plot in S2 Fig in S2 File is confirming this bias.

In order to estimate cardiovascular prevalence as an important underlying comorbidity, 27 articles were pooled and it was found 21% (16% - 27%) of fatal cases had this disease (Fig 4). High and significant heterogeneity was seen among included studies ($I^2$ = 88.67%, P<0.001). Publication bias was confirmed by funnel plot and Egger's test (t = 3.75, P = 0.001). S3 Fig in S2 File is shown these results. In order to conduct sensitivity analysis, two studies were excluded; but no significant changes had seen.

Pooled prevalence of kidney disease among death individuals with SARS-CoV-2 infection was estimated 7% (3% - 11%) (Fig 5). In random effect analysis significant heterogeneity was observed among the prevalence estimates of disease ($I^2$ = 82.89, P<0.001). The funnel plot of this analysis in S4 Fig in S2 File is not shown highly under reporting or publication bias (t = -0.62, P = 0.54)

The random effect meta-analysis revealed a pooled estimated of 8% (4%- 13%) for prevalence of COPD in COVID-19 died cases (Fig 6); however high heterogeneity was also a concern ($I^2$ = 74.19, P<0.001). Systemic pattern of funnel plot in S5 Fig is not shown publication bias, also these results were confirmed by Egger's test (t = 0.51, p = 0.62).

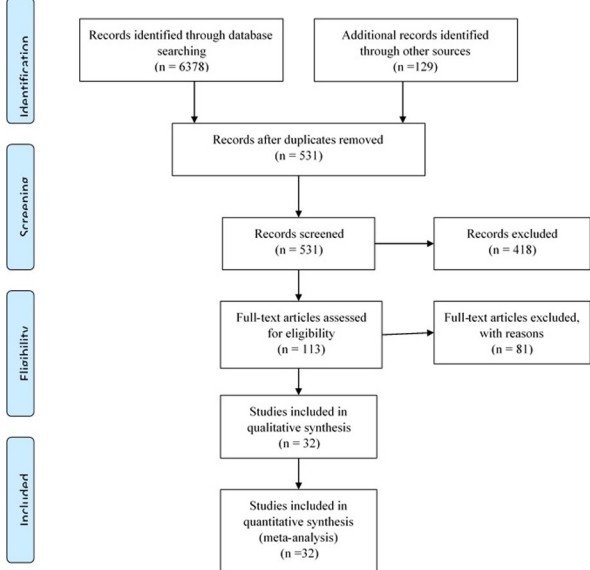

**Fig 1. PRISMA flow chart of the systematic literature review and article identification.**

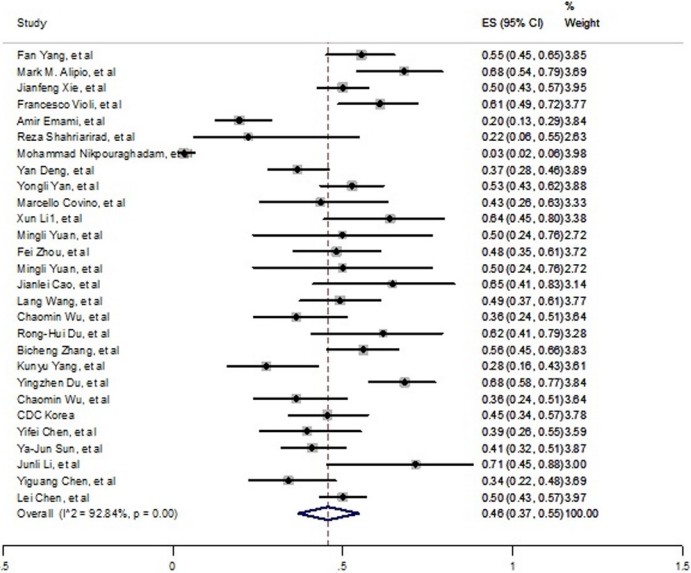

**Fig 2. Prevalence of hypertension among died patients with COVID-19.**

In order to evaluate pooled prevalence of malignancy in died cases of COVID-19, random effect analysis was done and the estimation 11% (4%-20%) was obtained with high and significant heterogeneity ($I^2$ = 95.82, P<0.001) which is shown in Fig 7. Sensitivity analysis reduce this estimation to 6% (3%-10%) and $I^2$ = 85.59% based on excluding Jianbo Tian's paper.

Based on fixed effect analysis, the forest plot drawn in Fig 8, and the pooled prevalence of liver disease in died cases related to COVID-19 was estimated 3% (2%- 6%) (Fig 8). Moreover, no publication bias was seen based on funnel plot and Egger's test (t = 0.57, p = 0.60, S7 Fig in S2 File).

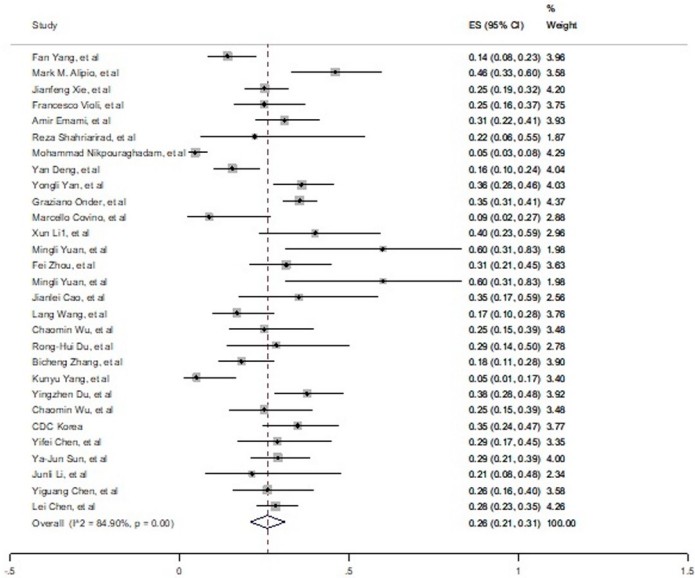

**Fig 3. Prevalence of diabetes among died patients with COVID-19.**

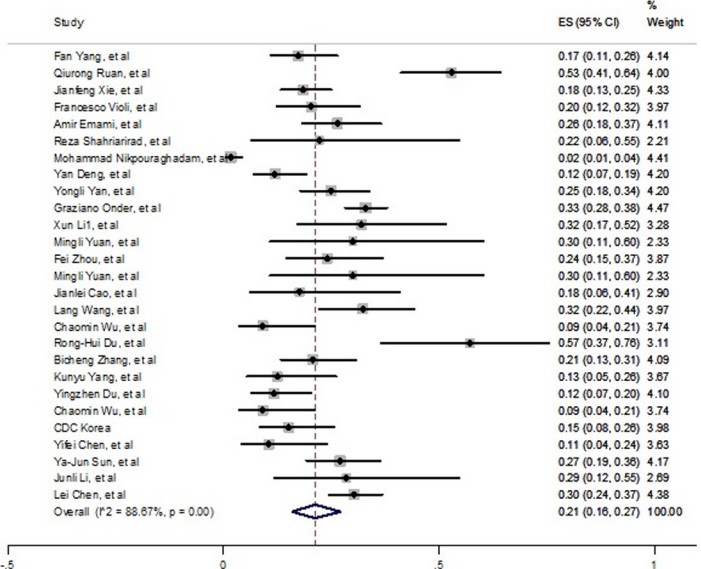

**Fig 4. Prevalence of cardiovascular disease among died patients with COVID-19.**

Lung disease as another one underlying disease were prevalent in died cases with SARS-CoV-2 infection. The pooled prevalence was 11% with confidence interval 95% (6% -18%) which is shown in Fig 9.

Heterogeneity was high between 4 included studies ($I^2$ = 62.45% P<0.001), but no publication was seen based on Egger's test and funnel plot (t = 1.26, P = 0.33, S8 Fig in S2 File).

Just two studies reported died cases from asthma with COVID-19. The combined results were estimated 9% (2%-19%).

By using the data from 13 included articles and fixed effect analysis, the prevalence of died cases with SARS-CoV-2 and cerebrovascular disease was estimated 12% (9%-15%). The related

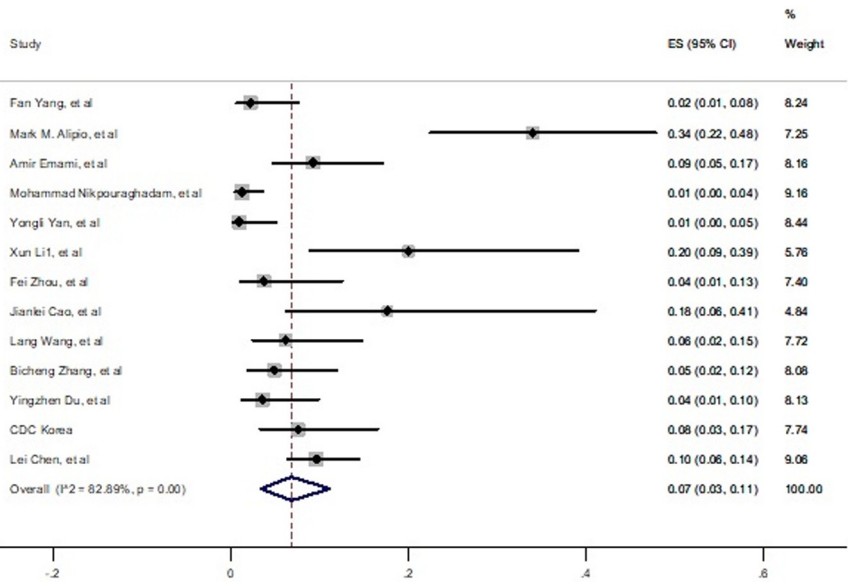

**Fig 5. Prevalence of kidney disease among died patients with COVID-19.**

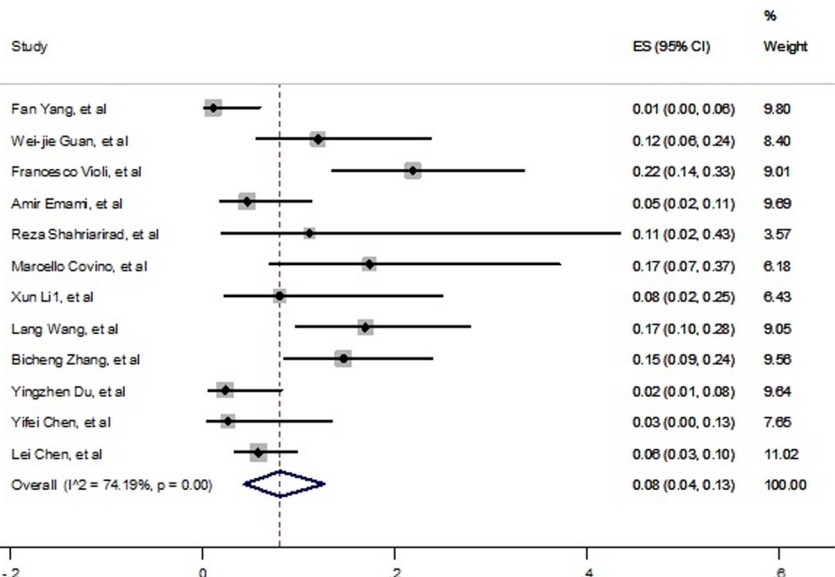

**Fig 6. Prevalence of COPD among died patients with COVID-19.**

forest plot is shown in Fig 10. Based on funnel plot and Egger's test, no publication bias was observed for these studies (t = 1.38, p = 0.19; S9 Fig in S2 File).

## Discussion

As pandemic progress, daily increase in active cases death related to SARS-CoV-2 infection become a global concern. Rapid distribution of COVID-19 all over the world, created a significant burden for health care systems. Various reasons like insufficient resources for several

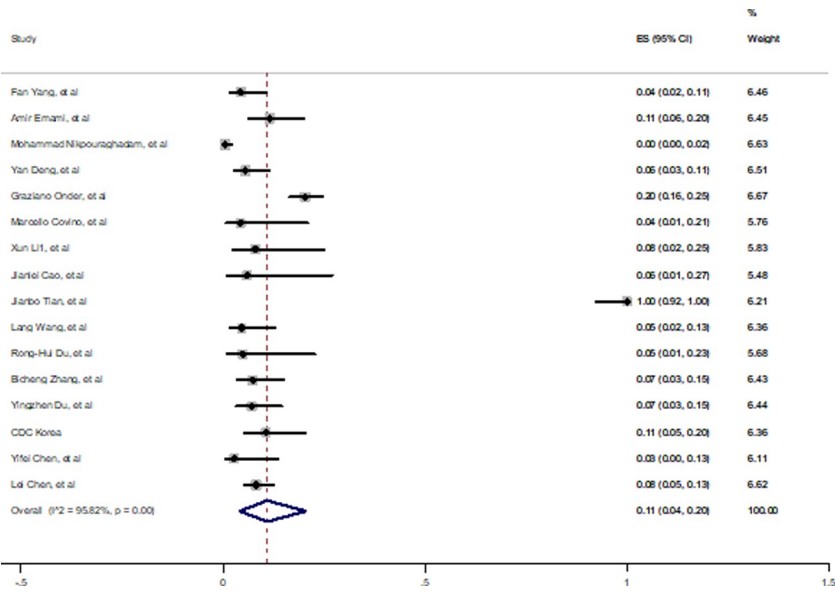

**Fig 7. Prevalence of malignancy among died patients with COVID-19.**

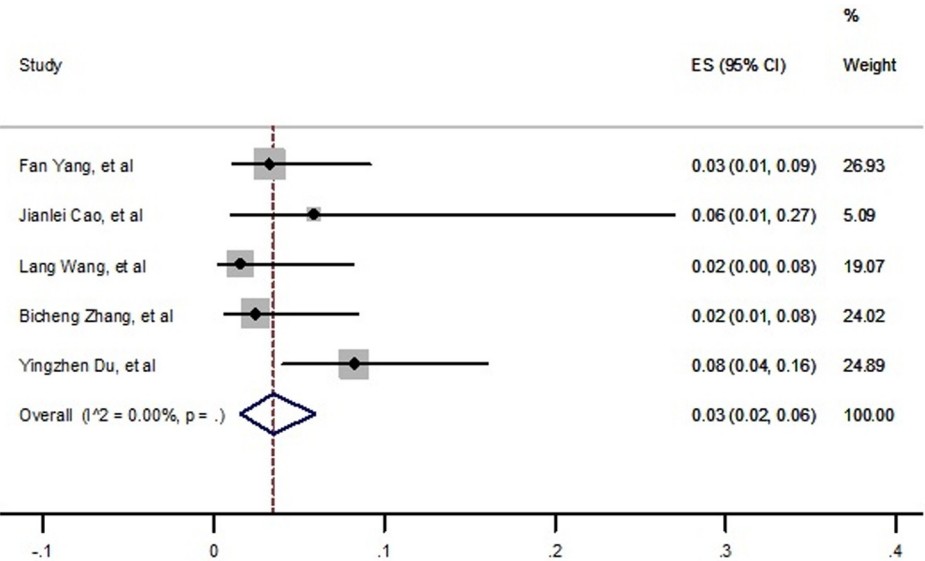

**Fig 8. Prevalence of malignancy among died patients with COVID-19.**

cases, prolong incubation time and the most important one; presence of comorbidities are known to be associated with high mortality rate [40, 41].

In the current study, we have conducted a systematic review and meta-analysis to identify the most prevalence underlying diseases in died cases related to SARS-CoV-2 infection. According to various published reports, it is proven that underlying diseases are associated with increased poor outcomes [5, 42]. Based on our results, the most hazardous comorbidities in fatal cases were hypertension, diabetes and cardiovascular diseases, respectively. Although the mechanism and severity of diseases and their poor outcome are unclear, but some reasons may justify this event. One of the possible explanations about high mortality in hypertensive

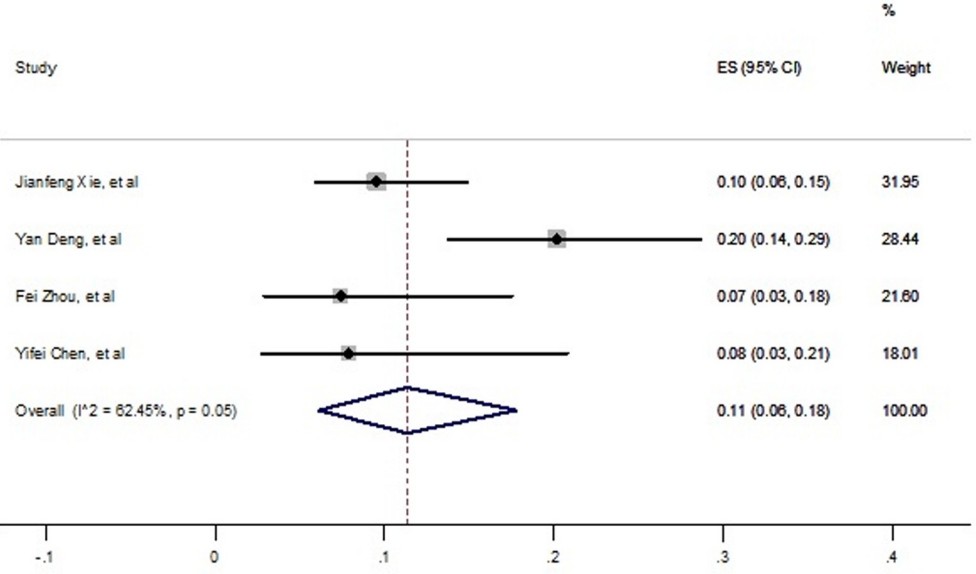

**Fig 9. Prevalence of lung disease among died patients with COVID-19.**

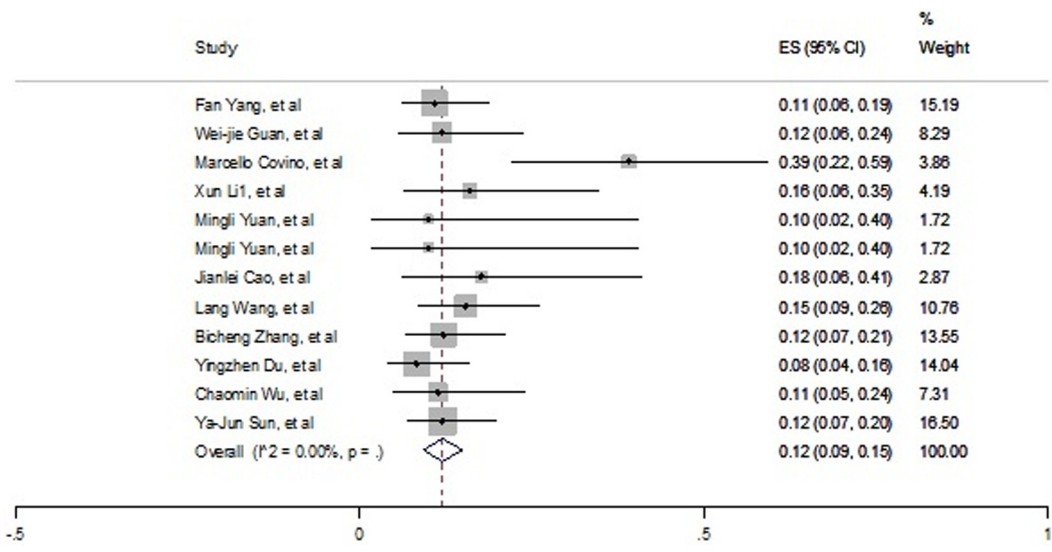

**Fig 10. Prevalence of cerebrovascular disease among died patients with COVID-19.**

and cardiac patients may be the function of ACE2 which may derive pulmonary hypertension and cardiovascular complications [43]. ACE2 has a critical role in immune and cardiovascular pathways. Since SARS-CoV-2 enter the cell and bind the ACE2 receptors, it is a major concern that may increase the risk of detrimental outcomes conferred by ACE inhibitors or ARBs [44–46]. Moreover, it is mentioned that morphologic and hemodynamic damage to heart tissues causes poor diagnosis in patients with COVID-19 and acute coronary syndrome [47].

Another important risk factor in fatal cases was diabetes. In order to justify the high prevalence of diabetic in fatal cases, it is suggested to evaluate the effect of SARS-CoV-2 on blood glucose which may related to ACE2. In previous SARS pandemic, it was found SARS-CoV-1, could cause hyperglycemia in people with no history of diabetes and it would persist almost 3 years after recovery and revealed temporary damage to beta cells [48]. It is suggested to follow blood glucose level in SARS-CoV-2 cases in acute stage. In other hands, the effect of anti-diabetic effect should not be ignored [49]. Increasing ACE2 expression and its relevance to COVID-19, causes that some researchers avoid or change some drugs (Thiazolidinedione) in these cases. However, impact of anti-diabetic drugs and their effects need more evaluation and researches to be clear [50].

Various studies about SARS-CoV-2 infection have shown patients with Chronic Kidney Disease (CKD) are vulnerable to be infected and become sever, since this novel virus enters the human body through ACE2 [51]. On the other hands, it has been documented that ACE2 expressions is high in kidney tissue either; so if patients with CKD become infected with SARS-CoV-2, their renal tubules maybe attacked in the first stage of infection. Moreover, SARS-CoV-2 may target the small arteries and capillaries in kidney, which in patients with history of CKD, this may cause rather impairment. Due to these reasons sever patients with CKD which need dialysis are more susceptible to the infection [41].

COVID-19 in malignant cases increase the risk of sever events and poor progress of disease. Chemotherapy, surgery, and treatment strategies cause these patients to become an immuno-suppressed population and vulnerable from the risk of infection [52].

Although the impact of SARS-CoV-2 in patients with liver disease or liver transplant is unclear, but due to the main viral receptor (ACE2) possible involvement of the liver and weak immune system, causes these patients to be more at risk of death in counterpart healthy

individuals. According to the date of searches for the current study, there was not found ant publication about acute chronic liver failure due to COVID-19.

Shortness of breath and cough are two symptoms which is seen in asthma patients and is seen in the COVID-19 cases, too. Based on the current analysis, 9% of fatal cases related to COVID-19 had history of asthma. There is no certain reason about the relation between mortality and asthma in patients with SARS-CoV-2 infection, but it recommended that nebulization should be avoided; since medical procedures may increase the risk of infection transmission [53].

According to the current results in this meta-analysis, 8% of patients with COPD comorbidities are in danger of rapid disease progression than their counterpart without COPD. In another systematic review which prevalence, severity and mortality associated with COPD and smoking in patients with COVID-19 were evaluated, it was found that the crude case fatality rate was 7.4% [54]. Also it was declared that this high mortality rate may be due to some co-existing of other comorbidities in these groups of patients [55].

Overall, it seems that drugs which used in underlying diseases aggravate COVID-19. In fact, corticosteroids, NSAIDS and some drugs acting on the renin-angiotensin system during the current pandemic which is in question and uncertainties [56]. Clinical impact of these treatments on COVID-19 infection needs more evaluation and should be clarified. Of limitation of current study could say high heterogeneity between studies in population and genetic limitation which most of the studies were from China.

## Conclusion

In summary, underlying diseases have a critical role in poor outcomes, severity of disease and high mortality rate related to COVID-19 cases. Patients with hypertension, cardiovascular disease and diabetes should be carefully monitored and be aware of health protocols.

## Supporting information

**S1 File. Search terms and quality assessment tables.**
(DOCX)

**S2 File. Funnel plot for publication bias assessment.**
(DOCX)

**S1 Checklist. PRISMA 2009 checklist.**
(DOC)

## Author Contributions

**Conceptualization:** Fatemeh Javanmardi.

**Data curation:** Fatemeh Javanmardi, Amir Emami.

**Formal analysis:** Fatemeh Javanmardi.

**Methodology:** Amir Emami.

**Project administration:** Amir Emami.

**Visualization:** Ali Akbari, Neda Pirbonyeh.

**Writing – original draft:** Fatemeh Javanmardi, Amir Emami.

**Writing – review & editing:** Abdolkhalegh Keshavarzi, Amir Emami.

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
