## [Decision Letter · Decision Letter 0]

1 Oct 2020

PONE-D-20-25586

Prevalence of Underlying Disease in Died cases of COVID-19: A systematic review and meta-analysis

PLOS ONE

Dear Dr. Emami,

Thank you for submitting your manuscript to PLOS ONE. After careful consideration, we feel that it has merit but does not fully meet PLOS ONE’s publication criteria as it currently stands. Therefore, we invite you to submit a revised version of the manuscript that addresses the points raised during the review process.

The reviewers have commented on your above paper. They have suggested that this manuscript be revised according to the reviewers suggestions and resubmitted.  Provided you address the changes recommended, the manuscript will be accepted for publication. 

We look forward to receiving your revised manuscript.

Kind regards,

Prof. Raffaele Serra, M.D., Ph.D

Academic Editor

PLOS ONE

Journal Requirements:

4. Please remove your figures from within your manuscript file, leaving only the individual TIFF/EPS image files, uploaded separately.  These will be automatically included in the reviewers’ PDF.

5. Please include captions for your Supporting Information files at the end of your manuscript, and update any in-text citations to match accordingly. Please see our Supporting Information guidelines for more information: http://journals.plos.org/plosone/s/supporting-information

Additional Editor Comments:

The manuscript is interesting and provided you address the changes recommended, the manuscript will be accepted for publication.

Reviewers' comments:

Reviewer's Responses to Questions

**Comments to the Author**

1. Is the manuscript technically sound, and do the data support the conclusions?

Reviewer #1: Yes

2. Has the statistical analysis been performed appropriately and rigorously? 

Reviewer #1: Yes

3. Have the authors made all data underlying the findings in their manuscript fully available?

Reviewer #1: Yes

4. Is the manuscript presented in an intelligible fashion and written in standard English?

Reviewer #1: Yes

5. Review Comments to the Author

Reviewer #1: The paper is very interesting and informative. I only suggest to deepen a little bit the issue of the relationship with cardiovascular disease in the discussion section. For example discuss and cite the following recent article on this topic: Ielapi N, et al. Cardiovascular disease as a biomarker for an increased risk of COVID-19 infection and related poor prognosis. Biomark Med. 2020 Jun;14(9):713-716.

6. PLOS authors have the option to publish the peer review history of their article (what does this mean?). If published, this will include your full peer review and any attached files.

Reviewer #1: No

---

## [Author Response · Author response to Decision Letter 0]

9 Oct 2020

Figure legend are provided in the end of article, after reference list. Also provide them as a separate word file in the attachments. 

Reference 10 is cited in the text and highlighted in GREEN Color

Revision 2: Thanks for your comments. I indicate the figure 10 citation in the main text and highlighted in GREEN color. 

Revision1: Thanks the reviewer for positive insight to our article.

Here I provide the point by point response to the comments. 

Editor Comments

Comments: please upload your study’s minimal underlying data set as either Supporting Information files or to a stable, public repository and include the relevant URLs, DOIs, or accession numbers within your revised cover letter. 

Answer: the current article is a systematic review and meta-analysis study. It does not have any original data. By this way I provide the table of included studies as a attached file with name Supporting Information.

Comments: Please amend either the title on the online submission form (via Edit Submission) or the title in the manuscript so that they are identical.

Answer: the title is “Prevalence of Underlying Diseases in died cases of COVID-19: a Systematic Review and Meta-Analysis”. It is identical in submission form and article file. 

Comments: Please remove your figures from within your manuscript file, leaving only the individual TIFF/EPS image files, uploaded separately. These will be automatically included in the reviewers’ PDF.

Answer: All the figures removed from the article and submitted separately. 

Comments: Please include captions for your Supporting Information files at the end of your manuscript, and update any in-text citations to match accordingly.

Answer: Thanks for your comment. I add the explanation of my supplementary file. 

Supplementary file1: Search terms and quality assessment tables

Supplementary file2: Funnel plot for publication bias assessment

Reviewers' comments:

Comments: The paper is very interesting and informative. I only suggest to deepen a little bit the issue of the relationship with cardiovascular disease in the discussion section. For example discuss and cite the following recent article on this topic: Ielapi N, et al. Cardiovascular disease as a biomarker for an increased risk of COVID-19 infection and related poor prognosis. Biomark Med. 2020 Jun;14(9):713-716.

Answer: Thanks for your suggestion. I add more explanation about cardiovascular disease in the discussion section and add the reference.

---

## [Editor Report · Decision Letter 1]

13 Oct 2020

Prevalence of Underlying Diseases in Died cases of COVID-19: A systematic review and meta-analysis

PONE-D-20-25586R1

Dear Dr. Emami,

We’re pleased to inform you that your manuscript has been judged scientifically suitable for publication and will be formally accepted for publication once it meets all outstanding technical requirements.

Kind regards,

Prof. Raffaele Serra, M.D., Ph.D

Academic Editor

PLOS ONE

Additional Editor Comments (optional):

amended manuscript is acceptable
---

## [Editor Report · Acceptance letter]

16 Oct 2020

PONE-D-20-25586R1 

Prevalence of Underlying Diseases in died cases of COVID-19: a Systematic Review and Meta-Analysis 

Dear Dr. Emami:

I'm pleased to inform you that your manuscript has been deemed suitable for publication in PLOS ONE. Congratulations! Your manuscript is now with our production department. 

Kind regards, 

on behalf of

Prof. Raffaele Serra 

Academic Editor

PLOS ONE